# Distributed Drive Electric Bus Handling Stability Control Based on Lyapunov Theory and Sliding Mode Control

**Feng Zhang** [1,*] **, Hongchao Xiao** [1] **, Yong Zhang** [1] **and Gang Gong** [2]

1 College of Mechanical Engineering and Automation, Huaqiao University, Xiamen 361021, China; xiaohongchao@stu.hqu.edu.cn (H.X.); zhangyong@hqu.edu.cn (Y.Z.)
2 Xiamen King Long United Automotive Industry Co., Ltd., Xiamen 361023, China; gongg@mail.king-long.com.cn
* Correspondence: zhangfeng@hqu.edu.cn

**Abstract:** To improve the handling stability of distributed drive electric buses, a vehicle stability control system based on direct yaw moment control (DYC) with a hierarchical control structure was designed. Considering that the vehicle dynamics system is highly nonlinear, a nonlinear controller based on Lyapunov stability theory was designed to calculate the required additional yaw moment of the vehicle in the upper controller. In the lower controller, the additional yaw moment is distributed to four wheel-side motors according to the equal proportion torque distribution method, and the direction of wheel-side motor output torque is determined based on the steering state of the vehicle. A co-simulation based on Simulink and Trucksim was conducted to verify the designed controller under two extreme conditions. Simulation results indicate that the proposed method performs feasibly and effectively in the handling stability of vehicles. Compared with traditional sliding mode control (SMC), the proposed control strategy can significantly reduce the chattering of the system, which provides a theoretical basis for the application of this yaw stability control method in engineering practice.

**Keywords:** distributed drive electric buses; DYC; Lyapunov theory

## 1. Introduction

Due to the advantages electric vehicles have in response speed, transmission efficiency, and simplified mechanical structure, the comprehensive performance of electric vehicles has significantly improved [1]. Generally, according to the different power drive forms, electric buses are divided into centralized drive and distributed drive [2]. Furthermore, the distributed drive more easily achieves complex motion and dynamics control, and it is also considered to be the ultimate drive form of an electric vehicle [3]. Distributed drive electric buses are driven by multiple independently controlled wheel-side motors, which have the characteristics of fast response and high energy efficiency [4]; however, the special power arrangement forms and structural configurations can lead to increased unsprung mass of full vehicles, which deteriorates the handling stability of the vehicle [5]. Since the rotational speed and torque of the wheel-side motor can be precisely controlled, some scholars and vehicle manufacturers apply the DYC method to raise the handling stability of vehicles [6–14]. A large number of studies have proven that the DYC method has a significant control effect of improving vehicle handling stability when the vehicle is steering rapidly or the road adhesion coefficient changes sharply [15–17].

DYC is a necessary active safety control system for vehicles, which typically adopts a hierarchy control structure [18–20]. The upper layer is the motion tracking control layer, which calculates the required additional yaw moment values based on the effective information of the DYC system, such as the desired state of a reference model, vehicle state feedback, driver input, etc. The lower layer is the torque distribution control layer, which distributes the additional yaw moment to each wheel-side motor to realize the control of

vehicle handling stability. Moreover, the upper layer controller can accurately calculate the required additional yaw moment value in real-time, which is critical to increasing the vehicle yaw stability. Motion tracking control methods commonly involve fuzzy control [21], fuzzy PID [22], traditional SMC [23], etc. Among them, fuzzy control has the advantage of not relying on an accurate model of the controlled object. Ma Haiying et al. applied the fuzzy controller to the DYC of nonlinear four-wheel steering vehicles and achieved a better simulation control effect [21]. The fuzzy rules are an essential component of fuzzy controllers, which are obtained through the expert experience and operation patterns of the operator. The rationality of fuzzy rules directly affects the performance of the controller, e.g., sometimes it declines the precision and efficiency of control. In fact, single fuzzy control can rarely achieve the desired control effect; in practice, fuzzy control is normally combined with other control algorithms to achieve better performance. Li Huimin et al. introduced fuzzy control to adaptively adjust PID parameters of different working conditions. The results indicated that this control method has better control performance and robustness [22]. Meanwhile, the control effect of the fuzzy controller sets higher requirements for selecting the membership function, which may affect the fuzzification effect of control inputs and outputs. Apart from that, a lot of scholars have employed SMC into DYC and achieved better control results [24–27]. The fundamental difference between SMC and other control methods is the discontinuity of control. In addition, to make the phase trajectory in SMC move along the predetermined sliding mode surface, a large switching gain function is required to constrain it, which will inevitably cause system chattering [28]. This chattering phenomenon diminishes the dynamic tracking performance of the control system. Some scholars have proposed solutions to the undesirable inherent chattering in SMC. Antonio and Basilio applied integral sliding mode control (ISMC), which effectively alleviates the chattering phenomenon of the system while maintaining the control effect [29]. Furthermore, Houzhong Zhang et al. employed adaptive fuzzy sliding mode control (FSMC) to raise the vehicle's handling stability more effectively under different working conditions; compared with traditional SMC, their control strategy effectively mitigates the chattering phenomenon of the control system [28].

Based on the summary of the research status of DYC, there is less research on DYC based on the characteristics of distributed drive electric buses. Since buses have a large mass, high center of mass, and their motion state is not prone to change, it is important to mitigate the chattering and abrupt change phenomenon of the yaw stability control system to raise the handling stability of electric buses; therefore, a new DYC strategy based on the Lyapunov stability theory is proposed. To diminish the steady-state error of the DYC system, the time integrator of yaw rate error is introduced into the Lyapunov function of yaw rate and sideslip angle. Our simulations prove that the designed control strategy can be employed in the DYC of distributed drive electric buses. Compared with traditional SMC, this control scheme can reduce the chattering and abrupt change of the system and improve the control effect of vehicle handling stability.

The rest of this paper is as follows: Initially, a 2-DOF reference model, a 7-DOF full vehicle dynamics model, and a wheel-side motor model are established. In addition, the design process of the direct yaw stability controller is described, involving the upper controller based on the Lyapunov stability theory and the torque distribution controller. Moreover, co-simulation experiments were carried out under two extreme conditions. The control process and control effect of the Lyapunov control scheme was analyzed and compared with SMC and with no control. Finally, the research results are summarized.

## 2. Vehicle Dynamics Model

### 2.1. Reference Model for Lateral Stability

A 2-DOF vehicle model is applied to calculate the ideal yaw rate $r_d$ and ideal sideslip angle $\beta_d$. When the actual yaw rate $r$ and sideslip angle $\beta$ trace ideal values, the yaw stability of the vehicle is in the ideal condition. As shown in Figure 1, this reference model takes the front wheel steering angle as input, and the front wheels and rear wheels are

subjected to lateral forces. Accordingly, the 2-DOF vehicle model mainly involves lateral and yaw motion.

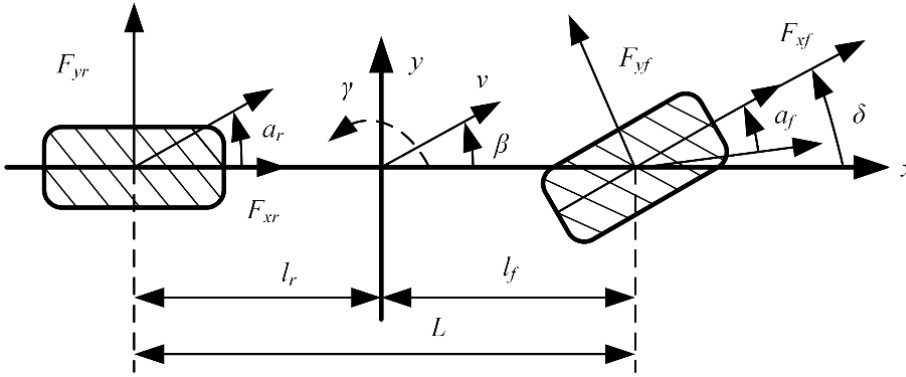

**Figure 1.** Vehicle linear 2-DOF model.

The lateral and yaw motion equations of the 2-DOF reference model are given as follows:

$$(C_f + C_r)\beta + \frac{1}{V_x}(l_f C_f - l_r C_r)r - C_f\delta = m(\dot{V}_y + V_x r) \tag{1}$$

$$(l_f C_f - l_r C_r)\beta + \frac{1}{V_x}(l_f^2 C_f + l_r^2 C_r)r - l_f C_f\delta = I_z\dot{r} \tag{2}$$

The definitions and values of the variables in Equations (1) and (2) and the entire paper are shown in Table 1. The 2-DOF vehicle model describes the relevance between the vehicle yaw stability, yaw rate, and sideslip angle in ideal conditions; thus, the desired yaw rate $r_d$ and the desired sideslip angle $\beta_d$ can be denoted in Equations (3) and (4) as:

$$r_d = \frac{V_x}{L + \frac{mV_x^2}{L}\left(\frac{l_f C_f - l_r C_r}{C_r C_f}\right)}\delta \tag{3}$$

$$\beta_d = \frac{2l_r(l_f + l_r)C_f C_r - mV_x^2 l_f C_f}{2l_r(l_f + l_r)^2 C_f C_r - mV_x^2(l_f C_f - l_r C_r)}\delta \tag{4}$$

**Table 1.** Vehicle parameters.

| Definition | Symbol | Value | Unit |
|---|---|---|---|
| Mass of vehicle | $m$ | 7360 | kg |
| Yaw moment of inertia | $I_z$ | 30,782.4 | kg·m$^2$ |
| Track width | $d_f, d_r$ | 2130 | mm |
| Distance from centroid to front axle | $l_f$ | 3100 | mm |
| Distance from centroid to rear axle | $l_r$ | 2900 | mm |
| Cornering stiffness of front tire | $C_f$ | 283,034 | N/rad |
| Cornering stiffness of rear tire | $C_r$ | 251,034 | N/rad |
| Wheel rolling radius | $R_e$ | 510 | mm |
| Longitude/Lateral speed | $V_x, V_y$ | | m/s |
| Height of centroid | $h_g$ | 1200 | mm |

Yaw rate $r$ and sideslip angle $\beta$ are restricted by road adhesion coefficient $\mu$; therefore, ideal yaw rate $r_d$ and ideal sideslip angle $\beta_d$ should be established boundary values relating to road adhesion conditions $\mu$. According to reference [30], the maximum value $r_{\max}$ of yaw rate is expressed in Equation (5).

$$r_d \leq |r_{\max}| = 0.85\frac{\mu g}{V_x} \tag{5}$$

Thus, the correction formula (6) for the ideal yaw rate $r_d$ is obtained from Equations (3) and (5),

$$r_{des} = \min\{|r_d|, |r_{\max}|\} \cdot \text{sgn}(r_d) \tag{6}$$

where the sgn (·) represents the sign function. Meanwhile, the expression of the maximum sideslip angle $\beta_{\max}$ can be acquired from Equations (3)–(5) [31],

$$\beta_{\max} = \arctan(0.02\mu g) \tag{7}$$

therefore, the correction formula (8) for ideal sideslip angle $\beta_d$ is given in Equation (8).

$$\beta_{des} = \min\{|\beta_d|, |\beta_{\max}|\} \cdot \text{sgn}(\beta_d) \tag{8}$$

### 2.2. 7-DOF Vehicle Dynamics Model

Since the handling stability of electric buses is primarily decided by lateral motion and yaw motion, a 7-DOF vehicle dynamics model with longitudinal, lateral, yaw, four-wheels rotational motion was established, as depicted in Figure 2. The 7-DOF model was applied to derive the yaw stability control algorithm.

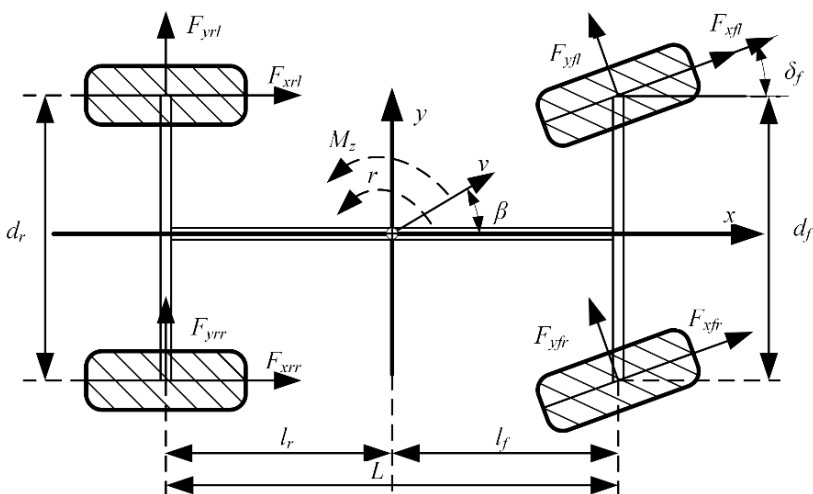

**Figure 2.** The 7-DOF vehicle model.

The longitudinal motion is expressed in Equation (9),

$$ma_x = m(\dot{V}_x - V_y r) = (F_{xfl} + F_{xfr})\cos\delta - (F_{yfl} + F_{yfr})\sin\delta + F_{xrl} + F_{xrr} \tag{9}$$

The lateral motion is shown in Equation (10),

$$ma_y = m(\dot{V}_y + V_x r) = (F_{yfr} + F_{yfl})\cos\delta + (F_{xfr} + F_{xfl})\sin\delta + F_{yrl} + F_{yrr} \tag{10}$$

Moreover, the yaw motion is given in Equation (11),

$$\begin{aligned} I_z \dot{r} = {} & l_f(F_{yfr} + F_{yfl})\cos\delta_f + l_f(F_{xfl} + F_{xfr})\sin\delta_f - l_r(F_{yrl} + F_{yrr}) \\ & -\tfrac{d_f}{2}(F_{yfr} - F_{yfl})\sin\delta_f + \tfrac{d_f}{2}(F_{xfr} - F_{xfl})\cos\delta_f + \tfrac{d_r}{2}(F_{xrr} - F_{xrl}) \end{aligned} \tag{11}$$

The rotational motion of each wheel is given in Equation (12),

$$J_w \dot{\omega}_w = T_{ij} - F_{xij} R_e \tag{12}$$

where $\delta_f$ is the front wheel steering angle, $F_{xij}$ is the longitudinal force on the tire, $F_{yij}$ represents the lateral force on the tire, $F_{zij}$ denotes the vertical force on the tire, $i$ represents the front and rear axles, $j$ represents the left and right wheels, $\omega_w$ represents the angular

rate of wheel, $T_{ij}$ refers to the driving or braking torque applied to wheels, and $J_w$ is the wheel inertia moment. In this paper, this 7-DOF model adopts the Magic tire model to provide tire forces, and the effects of suspension and aerodynamics are not considered.

### 2.3. Wheel-Side Motor Model

The distributed drive electric bus is driven by four independently controlled wheel-side motors. Since the transient response rate of modern motor drives is much faster than wheel, and considering the dominant pole of the closed-loop system, an electric motor drive model is simplified to a second-order transfer function [32]; its expression is as follows:

$$G(s) = \frac{T_a}{T_e} = \frac{1}{2\varepsilon^2 s^2 + 2\varepsilon s + 1} \tag{13}$$

where $T_a$ expresses the actual output torque of the wheel-side motor, $T_e$ shows the expected output torque of the wheel-side motor, and $\varepsilon$ denotes the damping ratio, which is related to the characteristics of the motor itself.

### 2.4. Full Vehicle Model Build in Trucksim

Considering that the vehicle motion state is complex, to increase the efficiency and accuracy of controller development, the full vehicle dynamics model was built in Trucksim. To realize the distributed drive scheme in Trucksim, the powertrain needs to be modified based on the original vehicle model. To begin with, the power transmission between the traditional transmission system and the wheel is interrupted by disconnecting all transmission links in front of the differential, and the output torque by the wheel-side motor is transmitted to the wheel through the wheel-side reducer; thus, the centralized drive form of the vehicle is transformed into a four-wheel-drive model. The vehicle structure parameters and dynamic parameters are given in Table 1.

## 3. Design Yaw Stability System

The designed direct yaw stability control system adopts a hierarchical control structure, as demonstrated in Figure 3. Among them, based on yaw rate error $e_r$ and sideslip angle error $e_\beta$, the Lyapunov controller is employed to calculate the required additional yaw moment to achieve the desired yaw stability. Based on the input of additional yaw moment, torque allocation strategy, and other constraints, the torque distribution controller translates the input of additional yaw moment into the output torque of each wheel-side motor; DYC can be achieved when each wheel is subjected to the corresponding driving or braking torque. In addition, the function of PID control is to make vehicles track desired motion states in the process of DYC implementation.

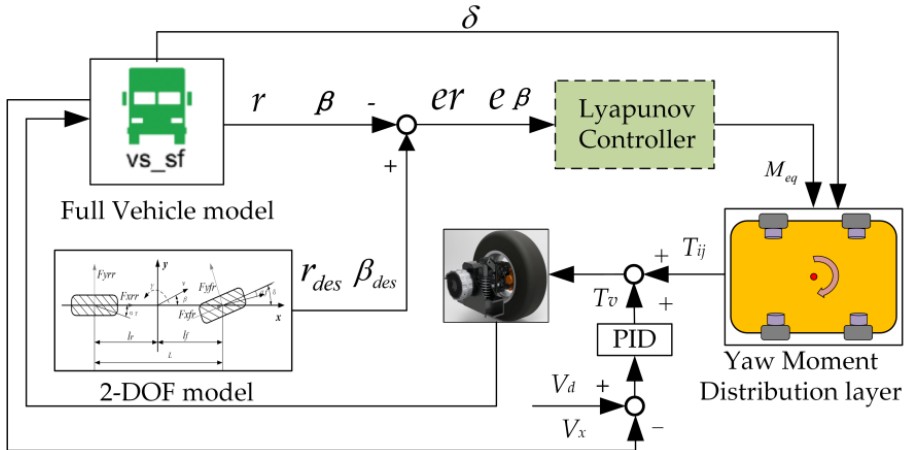

**Figure 3.** The yaw stability control structure.

### 3.1. Controller Based on The Lyapunov Theory

In this paper, the yaw rate $r$ and the sideslip angle $\beta$ are considered as the state variables of the yaw stability control system. To keep the handling stability of the vehicle in an ideal condition, the actual yaw rate $r$ and sideslip angle $\beta$ need to track the desired yaw rate $r_{des}$ and sideslip angle $\beta_{des}$, respectively, that is, the yaw rate error $e_r$ and the sideslip angle error $e_\beta$ should both be close to 0; therefore, the yaw rate errors $e_r$ and the sideslip angle errors $e_\beta$ are expressed as follows:

$$e_r = r - r_{des} \tag{14}$$

$$e_\beta = \beta - \beta_{des} \tag{15}$$

The corresponding Lyapunov function is given according to the Lyapunov stability theory. $V = s^2/2$ is a positive definite Lyapunov function, the time derivative $\dot{V} = s\dot{s}$ should be negative definite, then $\dot{s} = -\alpha s$, where $\alpha > 0$.

Define the equation $s$ about yaw rate $r$ and sideslip angle $\beta$, whose expression can be described by Equation (16).

$$s = k_1 e_\beta + k_2 e_r + k_3 \int e_r dt \tag{16}$$

where $k_1, k_2, k_3$ are the system control parameters, and $k_1 > 0, k_2 > 0, k_3 > 0$.

The time integrator of yaw rate error is introduced in Equation (16) to decrease the steady-state error of the system.

The derivative for both sides of Equation (16) is as follows:

$$\dot{s} = k_1 \dot{e}_\beta + k_2 \dot{e}_r + k_3 e_r = -\alpha(k_1 e_\beta + k_2 e_r + k_3 \int e_r dt) \tag{17}$$

According to Equation (11), we obtain:

$$I_z \dot{r} = l_f(F_{yfl} + F_{yfr}) \cos \delta_f + l_f(F_{xfl} + F_{xfr}) \sin \delta_f - l_r(F_{yrl} + F_{yrr}) - \frac{d_f}{2}(F_{yfr} - F_{yfl}) \sin \delta_f + M_{zc} \tag{18}$$

where $M_{zc}$ is the yaw moment applied by the yaw stability controller, and its expression is given by Equation (19).

$$M_{zc} = \frac{d_f}{2}(F_{xfr} - F_{xfl}) \cos \delta_f + \frac{d_r}{2}(F_{xrr} - F_{xrl}) \tag{19}$$

Thus, according to Equations (17) and (18), the expression for calculating the equivalent yaw moment $M_{eq}$ is as follows:

$$M_{eq} = I_z\left(\frac{1}{k_2}(-\alpha(k_1 e_\beta + k_2 e_r + k_3 \int e_r dt) - k_1 \dot{e}_\beta - k_3 e_r) + \dot{r}_{des}\right) - l_f(F_{yfl} + F_{yfr}) \cos \delta_f$$
$$+ l_r(F_{yrl} + F_{yrr}) - l_f(F_{xfl} - F_{xfr}) \sin \delta_f + \frac{d_f}{2}(F_{yfr} - F_{yfl}) \sin \delta_f \tag{20}$$

### 3.2. Torque Distribution Controller

According to the additional yaw moment, torque distribution method, and other constraints, the lower controller requires calculation of the output torque of each wheelside motor. Generally, because electric buses have a large mass, high center of mass, and their motion state is not prone to change, there is a large interaction force between their tires and the ground. To improve the stability margin of the wheels, during the implementation of DYC, the wheels should be subjected to the smallest additional longitudinal force possible. Furthermore, given the above characteristics of electric buses, when they run under extreme conditions, to ensure the handling stability and safety of electric buses, the response of the yaw stability control system should be fast; therefore, this paper adopts an equal proportion torque distribution strategy, that is, each wheel is applied the same magnitude of torque.

When the rolling resistance of tires is not considered, the magnitude of torque provided by each wheel-side motor is as follows:

$$T = mij \left| \frac{M_{eq}R_e}{2d\rho} \right| \tag{21}$$

where $p$ is the reduction ratio of wheel-side reducer, $d$ represents the front and rear track width, $d = d_f = d_r$, and the motor output torque is restricted by motor peak torque $T_{max}$ and road adhesion coefficient $\mu$. The correction formula of the wheel-side motor torque output is in Equation (22).

$$T_{mij} = \{ \left| \frac{M_{eq}R_e}{2d_i\rho} \right|, \frac{\mu F_{zij}R_e}{\rho}, T_{max} \} \tag{22}$$

After determining the torque magnitude, the lower controller is also required to decide the direction of the torque $T_{mij}$, that is, whether $T_{mij}$ is the driving torque or the braking torque. This requires judgment of the steering state of the vehicle based on the front wheel steering angle $\delta$ and the yaw rate error $e_r$, from which the direction of each wheel-side motor output torque $T_{mij}$ is determined. The torque distribution rules are shown in Table 2.

**Table 2.** The torque distribution rules.

| $\delta$ | $r-r_d$ | Steering State | $M_{eq}$ | Torque Distribution |
|---|---|---|---|---|
| $> 0$ | $r > 0$ | oversteer | $< 0$ | $T_{lf,lr} > 0,\ T_{rf,rr} < 0$ |
| $> 0$ | $r < 0$ | understeer | $> 0$ | $T_{lf,lr} < 0,\ T_{rf,rr} > 0$ |
| $< 0$ | $r > 0$ | understeer | $< 0$ | $T_{lf,lr} > 0,\ T_{rf,rr} < 0$ |
| $< 0$ | $r < 0$ | oversteer | $> 0$ | $T_{lf,lr} < 0,\ T_{rf,rr} > 0$ |

## 4. Results and Discussion

To prove the feasibility of the control scheme designed in this paper, the controller model based on the Lyapunov stability theory was developed in Simulink and the full vehicle model was established in Trucksim, and both were verified with co-simulations under the serpentine test and fishhook test, respectively. Furthermore, the control process and control effect of the Lyapunov control scheme were compared with SMC and with no control.

### 4.1. Serpentine Test

The serpentine test was mainly employed to examine the vehicle's yaw stability under emergency obstacle avoidance conditions. The simulation condition settings were as follows: the speed of vehicle was 80 km/h and the road adhesion coefficient was set to 0.5. The steering wheel angle input and simulation results are given in Figure 4.

Figure 4a shows the steering wheel angle input under the serpentine working conditions. Figure 4b,c show the yaw moments output by the sliding mode controller and Lyapunov controller, respectively. As illustrated in Figure 4b, in the time range of 0.8–1.5 s, 2.8–3.5 s, 4.8–5.5 s, and 6.8–7 s, respectively, the sliding mode control method shows sustained and large chattering phenomenon, and the largest chattering amplitude achieved was 25,000 N·m; however, in Figure 4c, the yaw moment output by the Lyapunov controller produced only small fluctuations during the whole test process. It can be deduced from the response curve of additional yaw moment that in emergency situations the Lyapunov control scheme produces a better handling stability control effect than the SMC method. On the other hand, the additional yaw moment output by the SMC controller shows an obvious chattering phenomenon, and when the system state changed sharply, the dynamic tracking control performance of the DYC system was reduced. The reason may be that the yaw moment required by the vehicle varied greatly before and after turning, and the sliding mode control scheme cannot adequately regulate the rate of change of the dynamic system. In addition, in the time ranges of 0–0.4 s, 1.6–2.1 s, 3.8–4.3 s, and 5.7–6.1 s, the

steering wheel angle varied in a smaller range, while the yaw moment calculated through the SMC control method had chattering with a small amplitude and high frequency. This indicates that the SMC control method is more sensitive to disturbances before reaching the slide mode surface. When the yaw moment is calculated, it needs to be distributed to the four wheel-side motors; the chattering phenomenon of yaw moment will inevitably cause motor torque fluctuation, which diminishes the comfort of the full vehicle and the service life of the wheel-side motor.

Figure 4d,e are the response curves of sideslip angle and yaw rate, respectively. As depicted in Figure 4d, the Lyapunov control method was applied to control the vehicle's sideslip angle within the range of −1.5° to 1.5°, and the response rate was fast. Compared to the vehicle with no control, although the SMC control method obviously reduces the range of the sideslip angle, at about 1.5 s, 3.5 s, and 5.5 s there is still an error of about 1° between the sideslip angle and its ideal value, indicating that the rear axle of the vehicle has a tendency to sideslip at these moments.

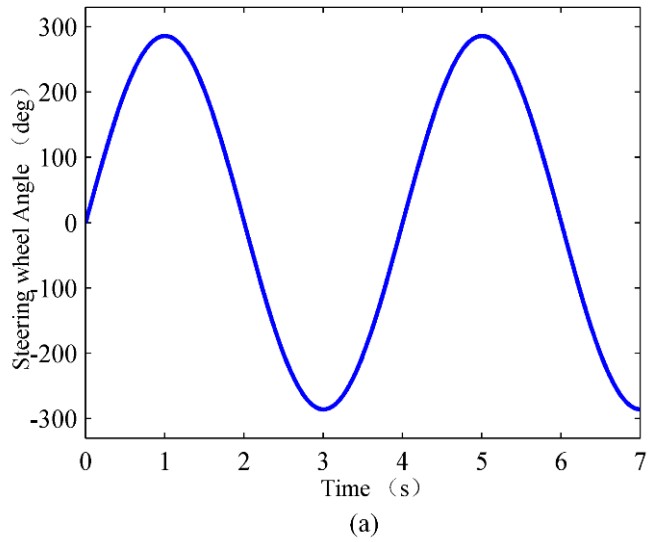

(a)

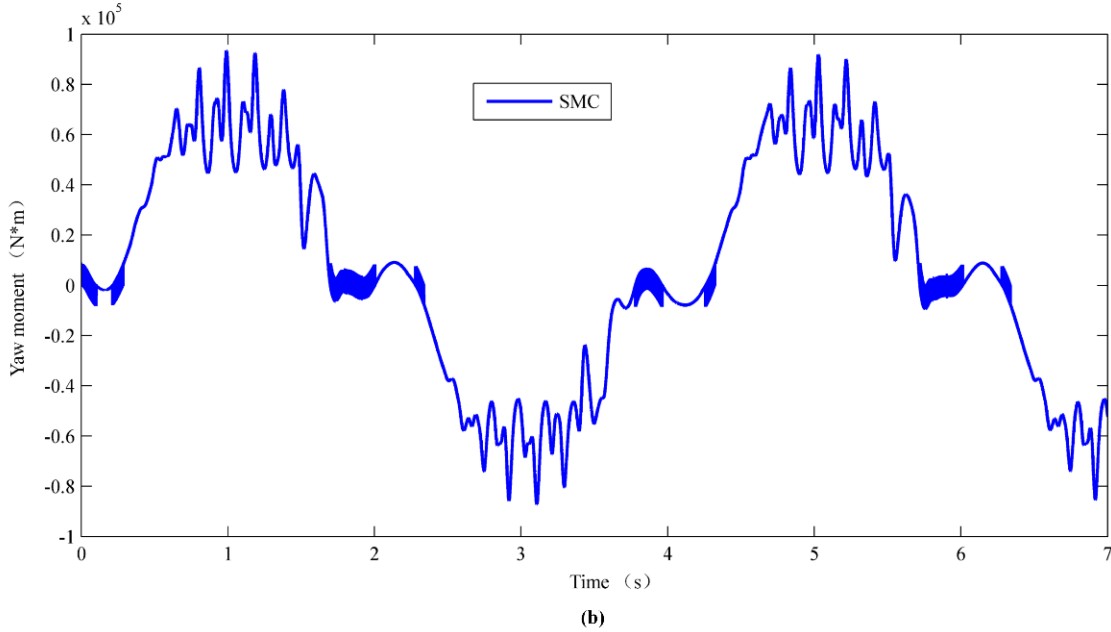

(b)

**Figure 4.** *Cont.*

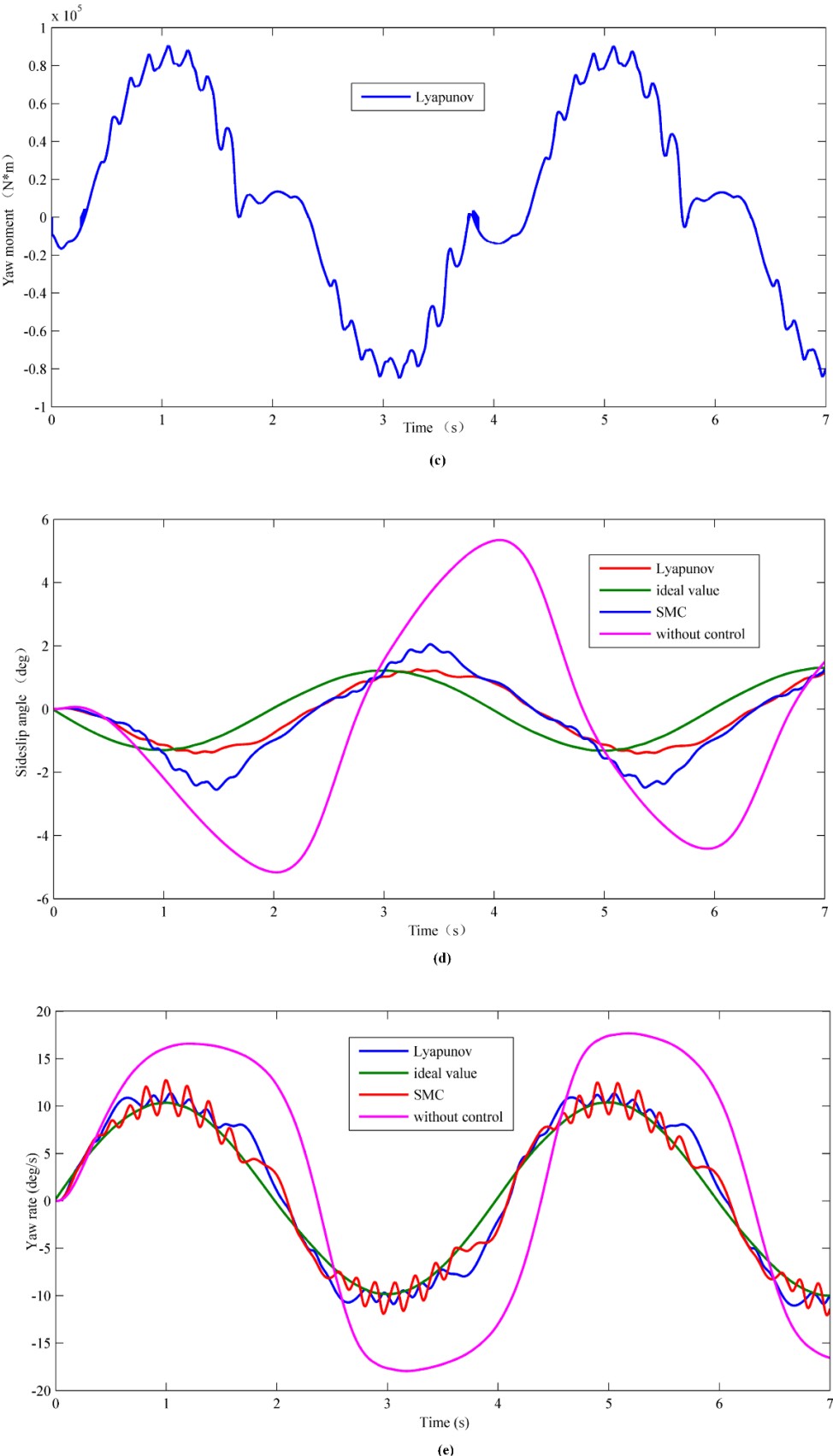

**Figure 4.** Serpentine test results: (**a**) steering wheel angle; (**b**) yaw moment response curve in SMC control mode; (**c**) yaw moment response curve in Lyapunov control method; (**d**) control results of sideslip angle; (**e**) control results of yaw rate.

Concerning the yaw rate, the SMC method and Lyapunov control strategy achieved an expected control effect: the yaw rate can track the ideal value change well during the whole test process, and it is controlled between $-12°/s$ to $12°/s$; however, under the without control action, the vehicle's yaw rate varied in the range of $-17°/s$ to $17°/s$, which seriously reduced the vehicle's handling stability control effect. Compared to the Lyapunov control strategy, in the time ranges of 0.8–1.5 s, 2.8–3.5 s, 4.8–5.5 s, and 6.8–7 s, the yaw rate of SMC had a larger chattering; this may have been caused by the chattering of the additional yaw moment.

Throughout the testing process, compared with SMC, the Lyapunov control method resulted in yaw rate and sideslip angle sufficiently tracking the change of ideal value, and the chattering of yaw rate and sideslip angle was alleviated, which can not only ensure vehicle handling stability but also improves vehicle comfort.

### 4.2. Fishhook Test

The fishhook test was used to examine the vehicle's handling stability under high-speed crash avoidance working conditions. This is a common method to test the active safety control system of the vehicle. The simulation working conditions set were as follows: the speed of the vehicle was 80 km/h and the road adhesion coefficient was 0.85; the steering wheel angle input and simulation control results are demonstrated in Figure 5.

Figure 5a shows the steering wheel angle input under the fishhook test. Figure 5b shows the additional yaw moment response curve output by the sliding mode controller and the Lyapunov controller. In Figure 5b, between 2 s to 3 s and 4 s to 7 s, the steering wheel angle input was large, the vehicle's motion state changed quickly during these two periods, the yaw moment output by sliding mode controller was chattering violently, and the average chattering amplitude achieved was 12,500 N·m. This chattering may have been caused by the existence of a large switching gain function in SMC. In addition, the yaw moment calculated by the SMC method shows an abrupt change of about 45,000 N·m at 3 s; however, during the entire test process, the Lyapunov control scheme effectively alleviated the chattering and abrupt change of yaw moment. It can be inferred that the Lyapunov controller has high control accuracy and effectively mitigates the system chattering and abrupt change caused by manually adjusting the switching gain function in SMC.

Comparing and analyzing the control effect of the Lyapunov control strategy, the SMC method, and with no control of the vehicle, Figure 5c–e show the sideslip angle, yaw rate, and trajectory response curve under the fishhook test, respectively. As indicated in Figure 5c, the Lyapunov control scheme was able to control the sideslip angle within the range of $-0.8°$ to $0.8°$, and it could track the ideal value of the sideslip angle well. Moreover, despite SMC restricting the vehicle sideslip angle to between $-1°$ and $1°$, in the time range of 2.2–3.2 s and 4.5–7 s, the deviation from the ideal value was still 0.2°; thus, the SMC control effect is inferior to the Lyapunov method. Additionally, from 3.2 s to 5 s, under uncontrolled conditions, the vehicle's sideslip angle changed from $-4°$ to 3.2°, and the change range of the sideslip angle was wide; this severely degrades vehicle handling stability and safety.

In Figure 5d, both the SMC method and Lyapunov control schemes achieved a desired control effect; the yaw rate was controlled between $-12°/s$ and $12°/s$, and it was able to track the ideal value change better. Compared with SMC, because the additional yaw moment calculated by the Lyapunov control method was relatively stable, the chattering of yaw rate was suppressed. In addition, when the vehicle was not controlled by DYC, its yaw rate varied greatly from 3.2 s to 4.2 s, and the change rate of yaw rate was about $-40°/s^2$; this will reduce the ride comfort and safety of a vehicle. Between 2 s to 3 s and 4 s to 7 s, when the vehicle was controlled by the SMC control method, its yaw rate showed obvious chattering, which reduced the yaw rate control accuracy.

Figure 5e shows the path trajectory diagram. When the vehicle was under uncontrolled action, due to the sideslip of the vehicle's rear axle, the vehicle appeared to oversteer, which caused the vehicle to deviate seriously from the desired trajectory. Although SMC could

improve the vehicle trajectory, because the chattering phenomenon reduced the dynamic tracking performance of the control system, it is slightly inferior to the designed Lyapunov control scheme in tracking the expected trajectory.

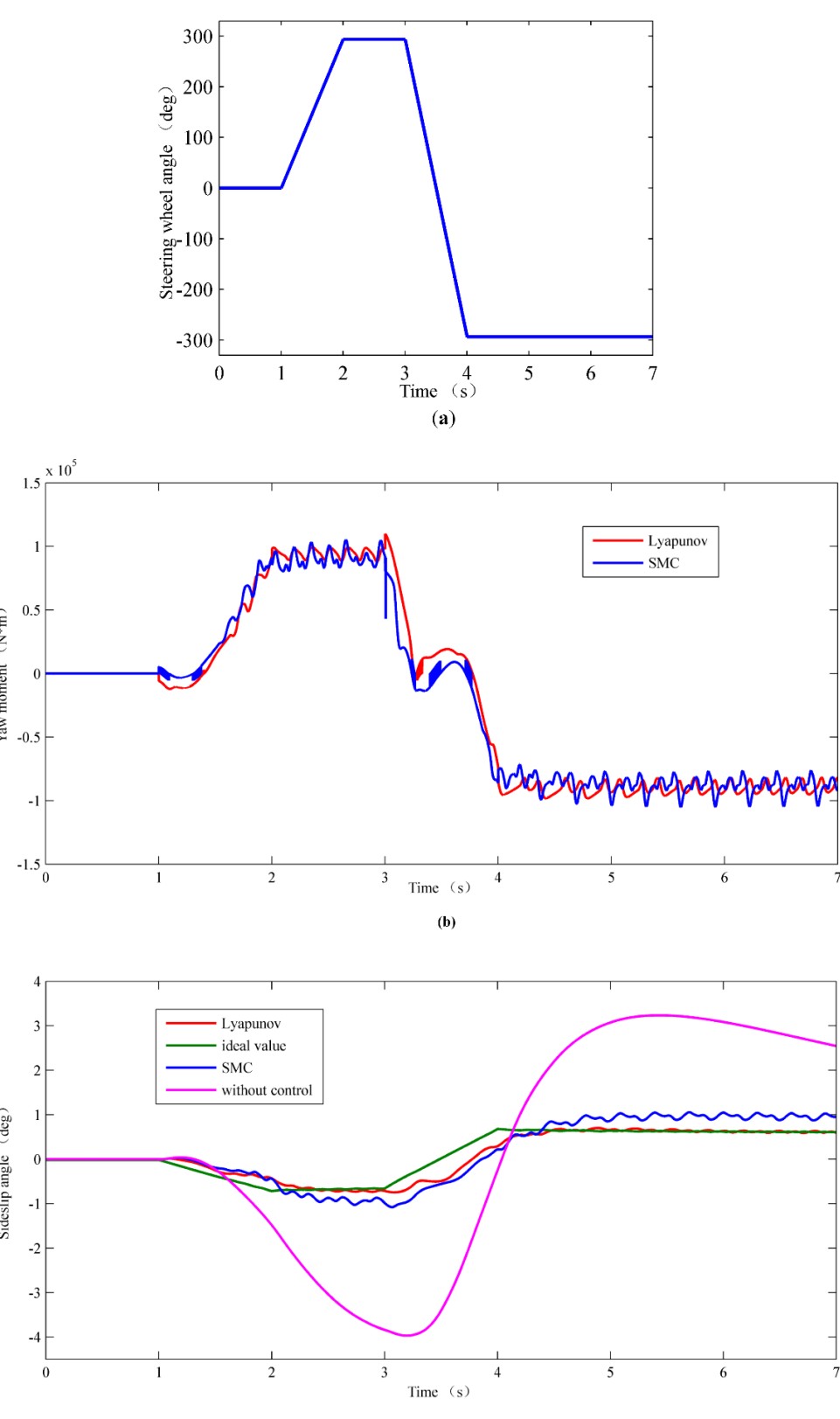

**Figure 5.** *Cont.*

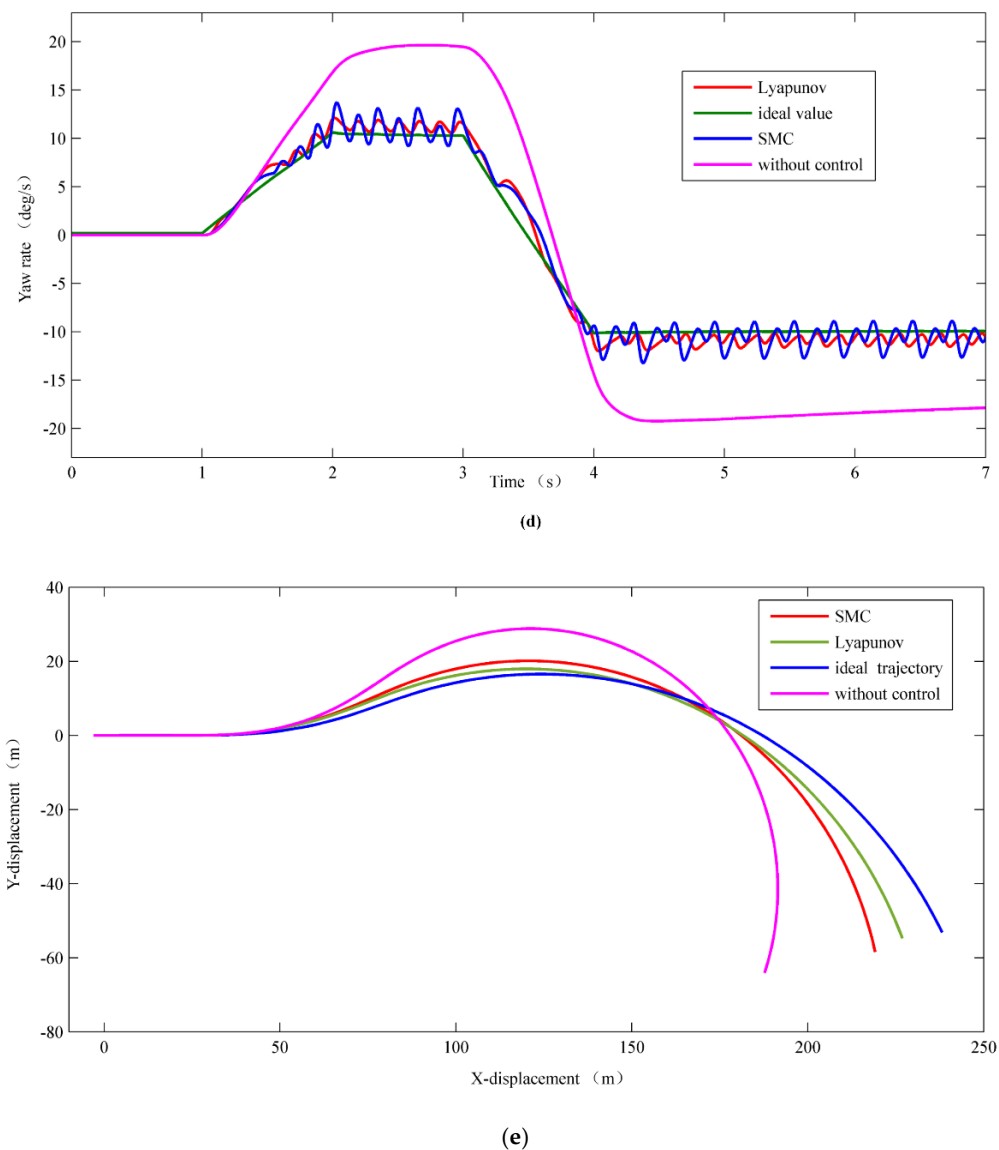

**Figure 5.** Fishhook test results: (**a**) steering wheel angle; (**b**) yaw moment response curve; (**c**) control results of sideslip angle; (**d**) control results of yaw rate; (**e**) path trajectory.

## 5. Conclusions

Due to the highly nonlinear features of full vehicle systems, vehicles are prone to instability under extreme conditions; thus, this paper presents a nonlinear DYC hierarchical control system based on Lyapunov stability theory. The study found that the Lyapunov control strategy has a better control effect on vehicle handling stability than SMC.

In terms of the control process, SMC is very sensitive to disturbance before reaching the sliding mode surface, and a small disturbance will cause a large control output. Furthermore, SMC has a manually adjustable switching gain function. As a result, the control system is prone to frequent chattering and large abrupt changes. The Lyapunov control strategy presented in this paper can avoid the disadvantages of SMC. It effectively alleviates the chattering and abrupt changes of the control system and also raises the control accuracy and dynamic tracking performance of the system.

**Author Contributions:** Conceived the original ideas, review, F.Z.; theoretical analysis, designed the controller, and writing—original draft preparation, H.X.; provided research methods, supervision, Y.Z.; software, performed the simulation analysis, G.G. All authors have read and agreed to the published version of the manuscript.

**Funding:** This research was supported by the Fundamental Research Funds for the Central Universities (ZQN-074).

**Institutional Review Board Statement:** Not applicable.

**Informed Consent Statement:** Not applicable.

**Data Availability Statement:** Not applicable.

**Acknowledgments:** We would like to thank Xiamen King Long United Automotive Industry Co. Ltd., especially Liang Su and Chao Chen, for helping the authors collect and organize literature and providing guidance to build the dynamic model.

**Conflicts of Interest:** The authors of this paper declare no conflict of interest.

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
