# Peer review of "Distributed Drive Electric Bus Handling Stability Control Based on Lyapunov Theory and Sliding Mode Control"

_actuators, doi:10.3390/act11030085_

Round 1
Reviewer 1 Report
The paper is interesting and shows sound contributions. It is well developed and results are well presented.However, the authors are advised to explore other control strategies mentioned in the literature. The following control strategies are recommended: Fuzzy Logic Controller, Artificial Neural Network, PID tuned by Genetic Algorithm . Other comments are shown in the attached file with the reviewer annotations.

Author Response
Dear reviewer, thank you for your comments. We have made modifications. Please check the attachment.

Reviewer 2 Report
In order to improve the handling stability of distributed drive electric bus, a direct yaw moment control strategy based on Lyapunov stability theory was proposed in this paper, and a vehicle stability control system based on the hierarchical control structure of direct yaw moment control was adopted. The effectiveness of the proposed control strategy was verified by comparing with that of traditional sliding mode control strategy. Overall, I found the work was interesting and of some practical significance. However, there are still some imperfections in the work. I hope that my comments would be useful for improving the quality of the paper. Some of detailed comments are as follows:
- It was mentioned on line 13 that “To improve the handling stability of distributed drive electric buses, a vehicle stability control system based on direct yaw moment control (DYC) with a hierarchical control structure is proposed.”. The improper expression of "proposed" is likely to lead to the misunderstanding that "vehicle stability control system based on the hierarchical control structure of direct yaw moment Control (DYC)" was proposed for the first time, which needs to be modified.
- It was mentioned on line 17 that “In the lower controller, the additional yaw moment is evenly distributed to four wheel-side motors according to the front wheel steering angle input and vehicle motion state parameter.”. Now that it is "evenly distributed", so it should be further explained why other parameters were still needed for additional yaw moment distribution.
- Real car experiments are lacking, and the results are not convincing enough.
- Recent references cited in the paper are not comprehensive enough and cannot reflect the author's professionalism in the field.
- It was mentioned on line 222 that “Therefore, this paper adopts equal proportion torque distribution strategy”. The reason for the “strategy” choice is not sufficient enough, which needs to be further explained.
Author Response

(The authors gave the same response as above.)

Reviewer 3 Report
Dear Authors,
minor comments are included in the PDF file.
There are two mandatory points, which should be addressed/commented:
- Why have you chosen the Lyapunov stability criterion? What other stability theories could be taken into account? Please, discuss this question.
- You haven't performed any experimental research. Could you please describe possible solutions: instrumentaiton, test procedures, safety matters for any experiment attempts that couuld be performed in order to verify your approach.
I've attached two files:
- the PDF file - your manuscript with my notes, not all of them are mandatory, some notes are my suggestions for your further research (notes starting with "M" are mandatory and you should address these comments);
- a Figure file, in which you'll find an illustration that correspondes to one of my comment in the PDF - manuscript file.
Good luck!

Author Response

(The authors gave the same response as above.)

Round 2
Reviewer 2 Report
Improvements have been made in the revised manuscript, and reasonable responses were given. The method proposed in the manuscript will be a good help for other researchers in related fields. However, further text editing is still needed, such as inconsistent index format of referenced literature. Then, the manuscript can be accepted.
Author Response
Dear editor, according to your suggestion, we have modified the format of references and cited several recent literatures. Thank you very much for your advice.
